# Risk Factors Associated with Preventable Hospitalisation among Rural Community-Dwelling Patients: A Systematic Review

**DOI:** 10.3390/ijerph192416487

**Published:** 2022-12-08

**Authors:** Andrew Ridge, Gregory M. Peterson, Rosie Nash

**Affiliations:** 1School of Pharmacy and Pharmacology, College of Health and Medicine, University of Tasmania, Hobart, TAS 7000, Australia; 2Huon Valley Health Centre, Huonville, TAS 7109, Australia; 3School of Medicine, College of Health and Medicine, University of Tasmania, Hobart, TAS 7000, Australia

**Keywords:** rural, primary care, preventable hospitalisation, risk factors, determinants, access, socioeconomic

## Abstract

Potentially preventable hospitalisations (PPHs) are common and increase the burden on already stretched healthcare services. Increasingly, psychosocial factors have been recognised as contributing to PPHs and these may be mitigated through greater attention to social capital. This systematic review investigates the factors associated with PPHs within rural populations. The review was designed, conducted, and reported according to PRISMA guidelines and registered with Prospero (ID: CRD42020152194). Four databases were systematically searched, and all potentially relevant papers were screened at the title/abstract level, followed by full-text review by at least two reviewers. Papers published between 2000–2022 were included. Quality assessment was conducted using Newcastle–Ottawa Scale and CASP Qualitative checklist. Of the thirteen papers included, eight were quantitative/descriptive and five were qualitative studies. All were from either Australia or the USA. Access to primary healthcare was frequently identified as a determinant of PPH. Socioeconomic, psychosocial, and geographical factors were commonly identified in the qualitative studies. This systematic review highlights the inherent attributes of rural populations that predispose them to PPHs. Equal importance should be given to supply/system factors that restrict access and patient-level factors that influence the ability and capacity of rural communities to receive appropriate primary healthcare.

## 1. Introduction

Potentially preventable hospitalisations (PPHs) occur when a medical condition that could have been avoided or managed with timely and adequate healthcare in the community results in a hospitalisation [1,2]. Other terms used interchangeably include potentially avoidable hospitalisation (PAH) [3], a hospitalisation due to an ambulatory care-sensitive condition (ACSC) [4], or an ambulatory care-sensitive hospitalisation (ACSH) [5]. Worldwide, these have a negative impact on health systems [6,7,8].

During 2020–2021, just over 672,000 public hospital admissions in Australia were classified as avoidable, equating to 5.7% of all admissions [9]. Admissions for potentially preventable reasons increased by 3.4% between 2012–2013 and 2017–2018 [10]; however, during the COVID-19 pandemic PPH rates decreased due to fewer vaccine-preventable hospital admissions [9]. Diabetes complications are the most frequent reason for PPHs in Australia [9]. Health systems in many western countries, including the USA and Australia, view preventing unnecessary hospital use as integral to maintaining an equitable, efficient, and sustainable health system [8,11]. In Australia, the National Healthcare Agreement uses PPHs as a performance indicator of primary and community health services to ensure the sustainability of the health system [12].

In Australia, 28% of the population live outside of major cities. Incrementally greater distances between these populations and services are reflected in the terms “rural, regional and remote” [13]. Rural populations often experience a unique combination of reduced access to healthcare services and socioeconomic, geographical, or systemic predispositions to avoidable hospitalisation. Access itself can be modulated by patient and population characteristics [14]. Internationally, rural populations are known to experience higher rates of PPHs and many potential targets for intervention have been suggested [15,16,17,18]. Rates of PPHs in Australia are lowest in major cities (22 per 1000) while regional and remote populations have higher rates (29 and 42 per 1000, respectively) [9,13]. The prevalence of chronic diseases and smoking, and health literacy challenges are known to be higher in rural populations [13]. Health system inequalities and the social, economic, and geographical characteristics of rural communities increase their burden of PPHs [19,20,21,22]. In contrast to urban contexts, however, there has been little synthesis of the determinants of PPHs within rural settings, which was the aim of this review.

## 2. Method

The research question was “what factors are associated with PPHs among rural community-dwelling patients?”. A protocol for the systematic review was developed according to the PRISMA-P guidelines [23] and registered with PROSPERO (ID CRD42020152194) [24].

Prior to commencement, the Cochrane Library [25] and PROSPERO [24] registry were searched for reviews underway or completed on the topic, and none were located. Four databases (MEDLINE, Web of Science, EMBASE and CINAHL) were searched according to the systematic search strategy described in Table A1 in Appendix A. Results from each database search were saved using EndNote X8 software [26] before being exported to Covidence [27] for duplicate removal and screening. A standardised data collection form was used to extract and collate information, including location of the study, definition of PPH used, range of independent variables assessed, and approach to statistical analysis.

The review team consisted of three researchers (AR, GP, and RN). Two reviewers independently screened the studies and excluded those that did not meet the inclusion criteria, firstly based on title and abstract, followed by a full-text reading. Studies were included when two reviewers agreed that the study met the inclusion criteria (Table 1). The third reviewer resolved conflicts where agreement could not be reached. Reasons for a paper being excluded at the full-text level were recorded.

Retained papers were independently assessed for quality by two research team members (AR and one other). The Newcastle–Ottawa Scale (NOS) for non-randomised studies [28] was used for quantitative studies, while qualitative studies were assessed with the CASP Qualitative Checklist [29,30]. A third reviewer (GP or RN) acted as adjudicator if agreement could not be reached by discussion. Because of the heterogenous nature of studies included, a narrative synthesis was used in the analysis [31].

## 3. Results

The initial search yielded 321 studies. After removal of duplicates (n = 166) and papers with titles or abstracts meeting exclusion criteria (n = 116), there were 39 papers remaining for full-text review. Exclusion of this number of studies was expected based on pilot searches and use of the very broad search terms. A further 26 papers were excluded for not meeting inclusion criteria, leaving 13 papers for review and analysis (Figure 1). Eight papers were quantitative/descriptive studies [32,33,34,35,36,37,38,39] and five were qualitative studies [40,41,42,43,44]. Only one study reported no statistically significant association between the variables studied (physician supply) and rural PPH risk [35].

### 3.1. Location

Descriptive studies were from either Australia (n = 3) or the USA (n = 5). The American studies were multi-state [35] or nation-wide [33,38], confined to one state (Nebraska) [39], or focused only on Indigenous Americans in rural California [34]. The Australian studies were set in Victoria [32], New South Wales [37], or Tasmania [36]. One study was excluded at the full-text review stage because, despite being conducted in a predominantly sparsely populated region of Spain, full-text review revealed that major cities were included in the analysis [45]. All the qualitative studies were set in Australia.

The population for study was variably described; this included being conducted in administrative areas serviced by “rural Indian Health Services” [34] or rural “health professional shortage areas” in Nebraska [39]. Other studies described a specific rural region of Tasmania [36,40,41], Victoria [32], or New South Wales [37,42,43,44] as the setting. The remainder used large, usually national, datasets from which rural patient data could be extracted.

All the qualitative studies in this review were from two research groups, each with their own geographical location of interest, namely the north coast of New South Wales [42,43,44] or Southern Tasmania [40,41].

### 3.2. Population

Children, incomplete data, or populations without access to hospitals were some of the exclusion criteria used in the studies (Table 1). An average age of patients in studies or presenting to hospital was not always reported; however, the proportions of people in age groups were often used to describe participants. The reported proportion of people aged >65 years ranged from 20.1% [34] to 49.5% [32]. Other studies reported an average age of participants between 63 years [36] and 75 years [38]. Females representativeness was between 48.6% [36] and 87.0% [42] in qualitative studies, whereas quantitative studies had a more equal distribution of sexes. The study by Korenbrot et al. was designed specifically to determine if there was a disparity in PPH rates between American Indians and Alaska Natives living in rural California [34].

### 3.3. Definitions and Data Sources

A variety of outcome definitions were used. Korenbrot et al. (2003) [34] used panel consensus to define the “avoidable” nature of admissions, while others used standard definitions of PPH [33,36,37,38].

All but one descriptive study [36] used minimum datasets linked with usage records, such as Medicare claims data [33,38] or national health databases [32,34,35,39]. Slimings et al. (2021) [37] used the Social Health Atlases of Australia in their ecological study, while Ridge et al. [36] linked hospital usage data with a general practice database. All qualitative papers used purposive sampling to recruit participants living in or servicing rural populations.

Characteristics defining rurality were taken from definitions issued by Australian Government organisations [32,36,37,40,41,43,44,46] or similar American bodies [33,35,39,47]. Korenbrot et al. included patients who were treated at rural Indian Health Services [34].

### 3.4. Analysis Methods

Logistic regression was used in several quantitative studies [32,36,39] to model PPH risk. Other similar statistical methods used included risk ratios [34], multivariate ordinary least squares regression [35], weighted negative binomial regression [33], and generalised linear modelling [37,38]. All the qualitative studies were interpreted using thematic analysis [40,41,42,43,44].

### 3.5. Predictors of PPH

Access

Restricted access to services was identified as a risk factor for PPH in most studies [32,33,35,38,42]. This included primary healthcare (PHC) physician supply or availability [32,33,36,39,40,42], the presence/absence of a dedicated PHC service in a rural area [38,39], or difficulty in navigating and optimising use of the healthcare system [42]. No further explanation of what was meant by “access to services” was reported by any study. Johnston et al. found that access to specialist care for rural patients, and *not* PHC physicians, was a driver of PPH [33], while only Laditka et al. found “no evidence that physician supply was associated with ACSH” in rural counties throughout America [35] (p.1161).

Factors related to location included remoteness [32,37,41] and transport [42]. Clinical associations with PPH included greater complexity of care needs [36,42] and overall comorbidity burden [36].

Psychosocial issues

Psychosocial issues linked to PPH included socioeconomic disadvantage or poverty [32,37,42], education or current occupation [32], isolation or living alone [36,41,43], health behaviours, beliefs or attitudes [36,41,42,44], and health literacy [36,41]. Racial or ethnic factors were confirmed or identified as a risk by Korenbrot et al. [34] and Slimings et al. [37]. Table 2 contains a summary of all data extracted from the 13 retained papers.

### 3.6. Quality Assessment

The quantitative papers scored highly for patient selection and outcome assessment criteria due to their use of large datasets; both patients with and without PPHs were taken from very representative datasets, and the predominantly retrospective approaches used ensured most outcomes were captured. Some papers scored less well for comparability criteria as only a limited number of potentially confounding factors were adequately controlled for in the statistical analyses; however, the findings of these studies are unlikely to be changed by this (Table 3).

The qualitative papers were assessed as generally having good validity, although the researchers’ role in collection and interpretation of data was often inadequately explained [41,43,44]. Several papers used pre-existing data from larger projects. The appropriateness of this data for new analyses and ethical considerations of re-using the data were not explicitly stated; however, the influence on the studies’ findings is probably insignificant [43,44] (Table 4).

## 4. Discussion

This systematic review brings together both descriptive and qualitative research, and thereby extends the understanding of risks associated with PPH. Specifically it sought to answer “what factors are associated with PPHs among rural community-dwelling patients?”.

Risks identified by this review are likely to be applicable to rural settings irrespective of nationality [48,49]. The effects of proximity to services, accessing the available healthcare workforce, and burden of socioeconomic disadvantage on PPH appear to be relevant to all rural settings. The disproportionate effect of lifestyle, behavioural, and environmental factors experienced by rural populations is a barrier to reducing health outcome disparities [50].

Poor access to PHC is an established risk factor contributing to PPH [51,52,53]. While a comparison of “access” in different healthcare systems is not always straightforward, it is evident from the current review that limited access is a common factor which predisposes rural populations to PPH. All but two papers suggested access to PHC services reduced PPH rates; Laditka et al. found no association between supply of PHC physician and PPH [35], while Johnston et al. showed that limited access to specialist medical, but *not* PHC, services was associated with PPH risk [33]. These findings are a reminder that, firstly, *supply* of PHC services is not always the most important determinant of PHC outcomes and, secondly, that the diversity of services available in rural areas should be governed by local community needs.

Penchansky et al. (1981) [54] previously described five dimensions (availability, accessibility, affordability, accommodation, and acceptability) of access which are evident in the 13 studies included in this review. The alignment of patient needs with the services offered by the healthcare system determines the ease, or difficulty, with which patients can receive healthcare. Notably, limited availability of an appropriate volume and range of services in rural communities was frequently identified as a risk for PPH [32,33,35,38,39,40,41,42,44]. Health workforce maldistribution is a known cause of lower numbers of health professionals serving rural areas [20]. Shortages in rural areas of general, and specialist, services limit patient access to PHC [33,55]. Collegial, social, and economic barriers to increasing the number of general practitioners (GPs) in rural Australia [56] contribute to GP visits per capita being half that of urban areas, with excessive wait times to see a GP in many rural settings [57]. Similar barriers to using primary healthcare have been observed in the USA [58]. Strategies beyond increasing the healthcare workforce and financial incentivisation of rural practice need to be considered to improve the volume and range of healthcare services for communities living in rural areas [57].

Accessibility (viz. geographical and transport barriers) and affordability of services were also identified in this review to be contributing to PPH amongst rural-dwelling patients. This finding could perhaps be a consequence of their easily quantifiable nature at an individual or population level. Providing and prioritising health services close to areas of high demand is an approach to resource allocation previously identified as crucial to reducing PPH [20].

Health system accommodation (the suitability of services to patients’ preferences) and acceptability in rural communities was demonstrated subjectively by patients’ preference to self-refer to hospitals for treatment and general satisfaction with local healthcare providers [40,41,44]. Providing different models of care, such as parallel provision of low-acuity and specialist services, could better accommodate patient preferences [59]. Involving under-utilised healthcare professionals, such as pharmacists and paramedics, has previously been suggested as a low-cost improvement to improving access [57].

An expanded understanding of access was offered by Levesque et al. (2013) [14], who identified five abilities that impact access. Initially, the ability to perceive, seek, reach, afford, and engage with healthcare services appear to mirror the domains identified by Penchansky et al. [54]. However, the qualitative papers reviewed here highlight how psychosocial factors influence “ability” and therefore access. Specific factors included health literacy [40,41], social determinants [40,41,42,43,46], and patient enablement [40,42]. Interestingly, all three factors are interconnected and exert their influence on each other and upon a person’s overall health across their entire life-course [60,61,62]. The social capital mechanisms [63] that improve or limit an individual’s access to healthcare were not explored in the studies included in this review, but provide an important focus for future research.

Disaggregating access into more nuanced components helps to shift the focus away from “supply and demand” thinking [14]. Understanding the non-clinical factors that influence a patient’s ability to interact with the health system could help to identify novel areas for interventions. Health literacy is lower in rural areas [21] and is known to be associated with poorer health outcomes [64,65], including a higher risk of hospitalisation [66,67]. A health-literate population would find it easier to access, understand and use healthcare information and services. It is important to understand that the health literacy strengths and challenges in rural communities are context-specific and may be different for each community. In order for healthcare services to be health literacy responsive [68] they must consider health literacy at both the community and individual level.

Related to health literacy, patient enablement is the “extent to which people understand their health conditions and have the confidence, skills, knowledge and ability to manage their health and wellbeing” [69]. Improving patient enablement ensures people can actively manage their own health, remain well, and avoid hospitalisations [70,71]. A passive approach to patient participation in healthcare was demonstrated in two of the qualitative studies included in this review [40,41], which is the direct opposite of patient enablement.

Influenced by a person’s social capital, the loneliness and social isolation they experience further limits an individual’s ability to harness support in times of need or to prevent ill health [72]. Otherwise referred to as distributed health literacy, a limited sphere of contact reduces the health literacy resources that can be used by patients, who themselves have low health literacy [73]. Compounding this further, isolation and loneliness can independently adversely affect their health [74,75]. Restrictions imposed during the COVID-19 pandemic highlighted the impact that isolation has on health outcomes [76], and provoked thought as to how isolated people can be reached and supported (e.g., telehealth) [76,77,78,79,80]. Reducing loneliness and social isolation may form part of a strategy to improve health literacy for individuals, communities, and organisations. In turn, this may reduce PPHs in rural communities. (Figure 2).

An emerging approach to addressing social factors, social prescribing, was mentioned in one paper [40]. This approach to holistic patient care provides a link between PHC and sources of support within the community [81]. The benefits of social prescribing are thought to be particularly important for rural populations [82]. This approach to reducing isolation and loneliness, while fostering enablement and building social capital [83], has increasing support at the policy level in Australia and overseas [84,85,86,87,88,89,90].

### Limitations

Only studies from Australia and the USA met the inclusion criteria and were included in the final analysis. The initial search results were reviewed and found to be accurate. Studies from several other countries were excluded for valid reasons (for example, Ingold et al. (2000) [91], Burgdorf et al. (2014) [92], Cloutier-Fisher et al. (2006) [93], O’Cathain et al. (2014) [94], Lynch et al. (2018) [95]). Further, the four qualitative papers included here were from two groups of researchers (both based in Australia). This may point to a need for greater emphasis on PPH research in rural settings, that is designed to capture stakeholder perceptions. These stakeholder insights are critical for informing policy and system improvements that meet local needs.

There is often a trade-off between producing statistically robust evidence and contextualisation of results; this was observed in the papers included in this review. Using large, highly linked data sets may identify prominent risks for PPH, but the applicability to local regions then needs careful consideration. Small quantitative studies, as always, are prone to possible biases and can be less applicable to other larger settings. For example, Ridge et al. (2021) [36] provides a useful snapshot of PPH patterns and risks in rural Tasmania. However, applying these findings to other rural areas in Australia or overseas ignores the specific geographic, demographic, socioeconomic, and healthcare system influences that are an integral part of each rural community.

## 5. Conclusions

This review is the first to highlight the importance of non-clinical determinants in contributing to PPH in rural communities and reinforces why elements within the access framework reported by Levesque et al. (2013) [14] should be considered. While poor “access” is a driver of PPH, considering factors beyond the “supply” of health services in rural areas is increasingly important. Patient-level attributes of health literacy, social isolation, and loneliness are important determinants of health. Rather than revisiting means of increasing healthcare provider numbers in rural areas, building the capacity of individuals, communities, and organisations to optimise their existing healthcare system is worthy of consideration. Employing a social capital approach to preventing PPHs may well be the answer.

## Figures and Tables

**Figure 1 ijerph-19-16487-f001:**
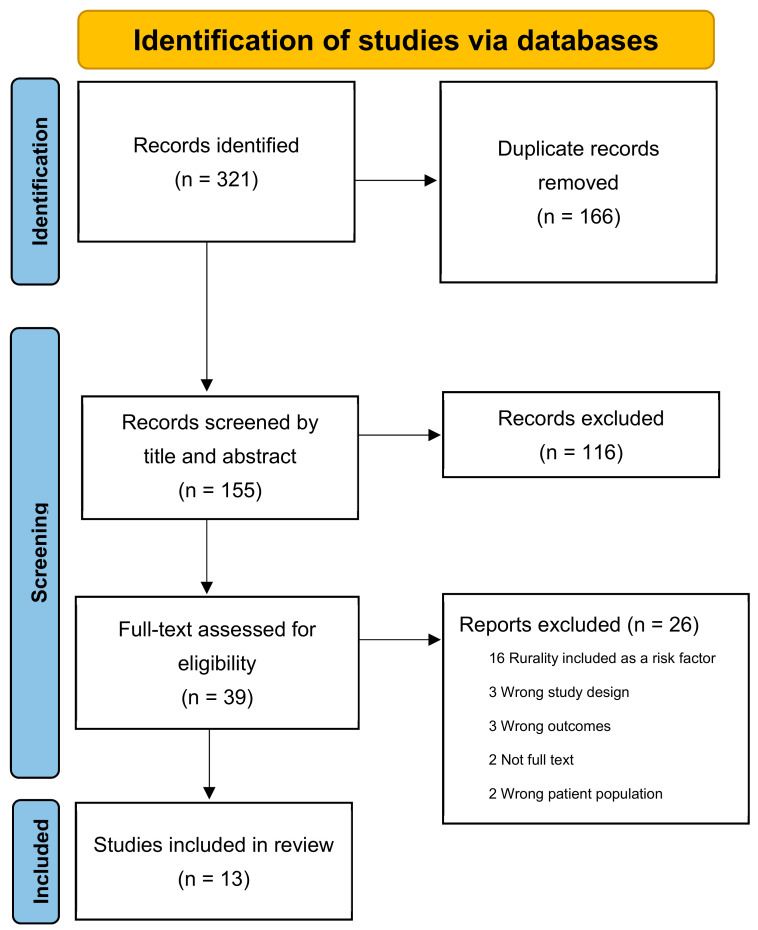
PRISMA flowchart for the review.

**Figure 2 ijerph-19-16487-f002:**
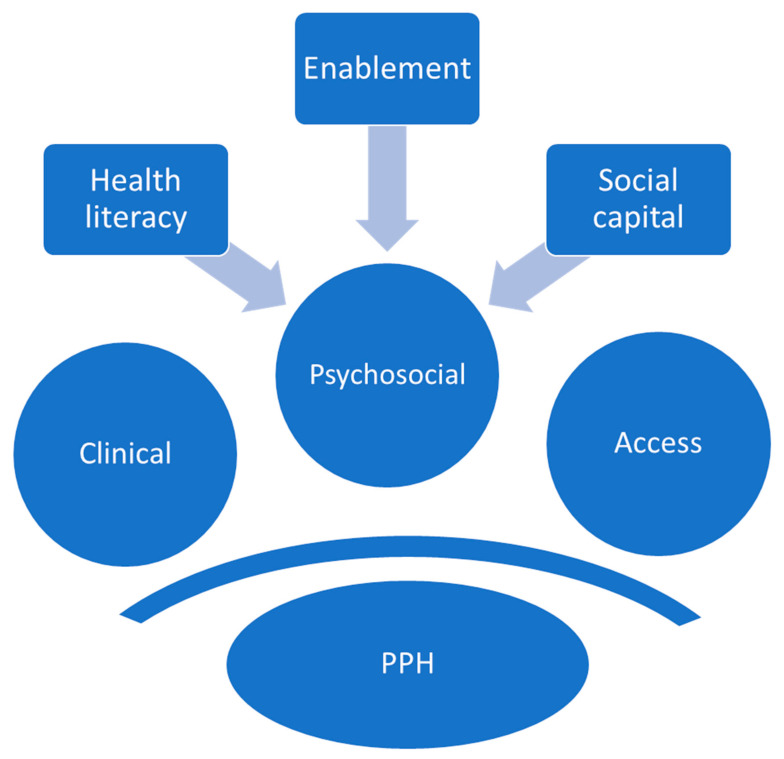
Key psychosocial factors that contribute to PPH for rural-dwelling patients.

**Table 1 ijerph-19-16487-t001:** Inclusion and exclusion criteria.

Inclusion Criteria	Exclusion Criteria
Participants/Population population representative of rural, community-dwelling patients, or providers of healthcare services in rural areas in developed economies at risk of a PPH. participants aged 18 years or over, or aged-based sub-groups of the population who are 18 years or over. Types of study to be included any original, full-text study where a factor, or factors, associated with potentially avoidable hospitalisation are identified and used to describe the risk of PPH in a rural population. Factors influencing PPH occurrence may be patient-level clinical, demographic, or social factors, or may reflect access to, or functioning of, the health system. set in rural areas or providing a comparison between rural and urban areas. published in the English language, and published after 1/1/2000	Types of studies to be excluded studies that identify factors associated with a repeat admission to hospital (i.e., a readmission within a specific timeframe). systematic reviews, meta-analyses, feasibility or pilot studies, case studies, correspondence, or conference/symposia abstracts. studies that include rurality as an independent variable in modelling (i.e., rurality was an investigated risk factor and not the outcome focus). studies based in residents of aged care facilities. studies investigating the impact of a single medication (or medication class) on PPH. studies in the context of one specific medical condition.

**Table 2 ijerph-19-16487-t002:** Description of papers included in the review.

Study (Year) Country	Study Type	Number of Participants–Data Source	Age	Sex (% Female)	Analysis Method	Key Findings
Korenbrot et al. (2003) [34] USA	Cross sectional	3920 California Department of Health Services	20.1% > 65 years (AI/AN group)	60.1	risk ratio with stratification by age, sex	AI/AN status increased PPH risk ratio for men (RR 2.26; 95%CI 1.39–2.98) & women (RR 1.87; 95%CI 1.46–3.17) Rates of PPH were only significantly higher for adult males aged > 45 years & women aged < 75 years after adjusting for age-group
Ansari et al. (2003) [32] Australia	Cross sectional	4,403,637 Victorian Admitted Episodes Dataset	42.0% aged > 65 years (ACSC group)	49.5	random effects multilevel regression	Strong associations with ACSH were observed for the following factors: insurance status reduces risk by 24% (ORadj = 0.76; 95%CI 0.75–0.77)) greatest remoteness increases risk by 17% (ORadj = 1.17; 95%CI 1.14–1.21) high population density reduces risk by 24% (ORadj = 0.76; 95%CI 0.74–0.78) greater GP presence reduces risk by 21% (ORadj = 0.79; 95%CI 0.78–0.81) more frequent GP visits reduces risk by 35% (ORadj = 0.65; 95%CI 0.64–0.67) the most disadvantaged socioeconomic areas had 40% higher risk (ORadj = 1.40; 95%CI 1.35–1.45) greater educational and occupational disadvantage increases risk by 48% (ORadj = 1.48; 95%CI 1.32–1.65) greater economic disadvantage increases risk by 56% (ORadj = 1.56; 95%CI 1.49–1.64)
Laditka et al. (2005) [35] USA	Cross sectional	948 counties Agency for Healthcare Research and Quality	n.s.	n.s.	multivariate ordinary least squares regression	Physician supply was not associated with ACSH in rural areas
Zhang et al. (2006) [39] USA	Cross sectional	538,580 Nebraska hospital discharge data (1999–2001)	n.s.	n.s.	multilevel logistic regression	Presence of ≥1 rural health clinic in rural areas was associated with a 5.5% reduction in risk of a ACSH due to a chronic disease among elderly patients (ORadj = 0.945; 95%CI 0.893–0.997)
Longman et al. (2011) [42] Australia	Qualitative	15 semi-structured interviews with healthcare providers	n.s.	86.7	thematic analysis	*External barriers influencing PPH risk*: complexity of services; availability, awareness, and ability to access services; greater care needs; patient poverty; rurality; transport *Internal barriers influencing PPH risk*: fear of change; “stoic” attitudes; difficulty accepting change in health status
Longman et al. (2013) [43] Australia	Qualitative	15 semi-structured interviews with healthcare providers	n.s.	n.s.	thematic analysis	Social isolation consistently identified as a risk factor for PPH among older patients with chronic diseases. Dimensions of social isolation included living alone, not socialising and being isolated from family
Johnston et al. (2019) [33] USA	Cross sectional	11,581 Centres for Medicare and Medicaid Services	n.s.	n.s.	weighted negative binomial regression	One or more specialist visits during the previous year was associated with a 15.9% lower preventable hospitalisation rate (explained 55% of the difference in preventable hospitalisation rates between rural and urban groups) Overall morbidity, heart failure (independently), lower income and being unmarried were all associated with a higher preventable hospitalisation risk.
Ridge et al. (2021) [41] Australia	Qualitative	14 semi-structured interviews with healthcare providers	n.s.	n.s.	thematic analysis	Health literacy challenges; access to PHC; perceived convenience of hospital treatment
Longman et al. (2021) [44] Australia	Qualitative	148 preventable admissions reviewed by expert panel	n.s.	n.s.	thematic analysis	*System issues*: community-based services inadequate or not referred to; poor connections between services; problems with specialist services *Clinician issues*: GP care inadequate *Patient issues:* adherence/self-management; patient’s engagement with existing services
Slimings et al. (2021) [37] Australia	Ecological	89 LGAs Social Health Atlases of Australia	20.8% aged > 65 years	49.80	multivariable analysis using generalised linear model	Remoteness, Indigenous percentage, and socioeconomic disadvantage were independently associated with preventable hospitalisation in rural NSW. Socioeconomic factors (measured by internet access) and Indigenous percentage remained significant in the adjusted model with 416.5 fewer (95%CI−597.6-−235.5; *p* <0.001) and 367.0 (95%CI 68.8–665.2; *p* = 0.041) more preventable hospitalisations per 100,000 population, respectively, between 2013–2017.
Ridge et al. (2021) [36] Australia	Cross sectional	436 Admitted Patient Data Collection dataset	62.5 years	48.6	multivariate logistic regression	Being single/unmarried (OR 2.43; 95%CI 1.38–4.28), greater comorbidity burden (as measured by higher Charlson Comorbidity Index scores) (OR 1.40; 95%CI 1.13–1.74) and number of general practice visits in the preceding 12 months (OR 1.09, 95%CI 1.05–1.14) were all associated with a higher risk of PPH
Ridge et al. (2022) [40] Australia	Qualitative	10 semi-structured patient interviews	68 years (range 47–91)	40.0	thematic analysis	Patient self-efficacy and health literacy; community support networks; access to PHC services
Wright et al. (2022) [38] USA	Cross sectional	8,483,758 person-year observations Medicare claims & Master Beneficiary Summary File (2012–2018)	75.1 years (IQR 69–80)	68.3	negative binomial and linear probability models	Dual-registered persons in rural areas receiving 100% of their PHC at a FQHC demonstrated a lower propensity for ACSH (marginal effect 0.3%; 95%CI 0.1–0.4)

ACSC = ambulatory care sensitive condition; ACSH = ambulatory care sensitive hospitalisation; AHPF = Australian Health Performance Framework; AHRQ = Agency for Healthcare Research and Quality; AI/AN = American Indians/Alaska Natives; FQHC = Federally Qualified Health Centre; GP = general practitioner; IQR = interquartile range; LGA = Local Government Area; n.s. = not stated; NSW = New South Wales; ORadj = adjusted odds ratio; PHC = primary healthcare; PPH = Potentially preventable hospitalisation; RR = risk ratio.

**Table 3 ijerph-19-16487-t003:** Newcastle–Ottawa Scores for quantitative papers.

	Korenbrot et al. (2003) [34]	Ansari et al. (2003) [32]	Laditka et al. (2005) [35]	Zhang et al. (2006) [39]	Johnston et al. (2019) [33]	Slimings et al. (2021) [37]	Ridge et al. (2021) [36]	Wright et al. (2022) [38]
Selection	4	4	4	4	4	4	4	4
Comparability	1	2	2	2	2	1	2	1
Outcome	3	3	3	3	3	3	3	3

nb: score based on the number of Newcastle-Ottawa Score domains that are met, with a maximum score of 9 possible across the three items (‘selection’, ‘comparability’ and ‘outcome’) [27].

**Table 4 ijerph-19-16487-t004:** CASP rating for qualitative papers.

	Longman et al. (2011) [42]	Longman et al. (2013) [43]	Longman et al. (2021) [44]	Ridge et al. (2021) [41]	Ridge et al. (2022) [40]
Section A: Are the results valid?					
1. Was there a clear statement of the aims of the research?	✓	✓	✓	✓	✓
2. Is a qualitative methodology appropriate?	✓	✓	✓	✓	✓
3. Was the research design appropriate to address the aims of the research?	✓	?	✓	✓	✓
4. Was the recruitment strategy appropriate to the aims of the research?	✓	?	✓	✓	✓
5. Was the data collected in a way that addressed the research issue?	✓	?	✓	✓	✓
6. Has the relationship between researcher and participants been adequately considered?	*x*	*x*	?	?	✓
Section B: What are the results?					
7. Have ethical issues been taken into consideration?	✓	?	?	✓	✓
8. Was the data analysis sufficiently rigorous?	*x*	?	✓	✓	✓
9. Is there a clear statement of findings?	✓	✓	✓	✓	✓
Section C: Will the results help locally?					
10. How valuable is the research?	7 ✓ 2 *x*	3 ✓ 5 ? 1 *x*	7 ✓ 2 ?	8 ✓ 1 ?	9 ✓

CASP scoring as per Critical Appraisal Skills Programme (2018) b [29]: ✓-yes, ?–cannot tell, *x*-no.

## Data Availability

Not applicable.

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
