# Peer review of "Risk Factors Associated with Preventable Hospitalisation among Rural Community-Dwelling Patients: A Systematic Review"

_ijerph, 2022, doi:10.3390/ijerph192416487_

Round 1

Reviewer 1 Report

It is an attempt on community care to find out the contributing risk factors for PPH in rural communities. It is a meaningful systematic review research to discuss the risk factors from different dimenion (such as individual and services) associated with PPH in rural community.

Thanks for the opportunity to peer review.

The purpose of the research is clear and consistent. Comments and suggestions are below as following

1.The authors did not describe the acronym for ACSH, IQR in footnote of table 2.

2.The acronym for AH and NHA have described in the footnote in table 2. However, those words were not shown in Table 2.

3. Please check the correctness of the number (416.5) for the key findings of Slimings and Moore [37] Australia in table 2 “….. in the adjusted model with 416.5 (95%CI −597.6-−235.5; p <.001) and …...,

4.Could the authors describe clearly the risk of bias about the detailed items of Newcastle-Ottawa Scores in table 3?

Reviewer 2 Report

This manuscript fully addresses this research topic, with the only recommendation that since more than half of the articles were excluded from duplicates, the authors must be more explicit in their search strategy to explain why this phenomenon occurs.

Reviewer 3 Report

First of all, congratulations to the authors for their work. As the only aspect to modify, I would suggest that in the introduction more information should be included and the paragraphs should be better linked between them. 
